# Validity and Reliability of Inertial Measurement Units in Active Range of Motion Assessment in the Hip Joint

**DOI:** 10.3390/s23218782

**Published:** 2023-10-28

**Authors:** Łukasz Stołowski, Maciej Niedziela, Bartłomiej Lubiatowski, Przemysław Lubiatowski, Tomasz Piontek

**Affiliations:** 1Department of Orthopedic Surgery, Rehasport Clinic, 60-201 Poznan, Poland; maciej.niedziela@rehasport.pl (M.N.); p.lubiatowski@rehasport.pl (P.L.); tomasz.piontek@rehasport.pl (T.P.); 2Doctoral School, Poznan University of Medical Sciences, 60-812 Poznan, Poland; 3RSQ Technologies, 27 Grudnia 3, 61-737 Poznan, Poland; bl@rsqtechnologies.com; 4Orthopaedics, Traumatology and Hand Surgery Department, Poznan University of Medical Sciences, 28 Czerwca 1956, No. 135/147, 61-545 Poznan, Poland; 5Department of Spine Disorders and Pediatric Orthopedics, University of Medical Sciences Poznan, 61-701 Poznan, Poland

**Keywords:** IMU, hip range of motion, goniometer, inclinometer

## Abstract

Measuring joint range of motion has traditionally occurred with a universal goniometer, inclinometer, or expensive laboratory systems. The popularity of the inertial measurement unit (IMU) is growing, enabling the small and even single portable device to measure the range of motion. Until now, they were not used to assess hip joint range of motion. Our study aimed to check the validity of IMUs in assessing hip range of motion and compare them to other measurement devices—universal goniometer and inclinometer. Twenty participants completed three hip movements (flexion in standing and prone internal and external rotation) on both hips. Two testers independently assessed each movement with a goniometer, digital inclinometer, and IMU at different time points. To assess the agreement of active hip ROM between devices, Intraclass Correlation Coefficient (ICC) and Bland–Altman analysis were used. Furthermore, inter-rater and intra-rater reliability were also assessed by using ICC and Bland–Altman analysis. Limits of agreement (LOA) were calculated using Bland–Altman plots. The IMU demonstrated good to excellent validity (ICC: 0.87–0.99) compared to the goniometer and digital inclinometer, with LOAs < 9°, across all tested movements. Intra-rater reliability was excellent for all devices (ICC: 0.87–0.99) with LOAs < 7°. However, inter-rater reliability was moderate for flexion (ICC: 0.58–0.59, LOAs < 22.4) and poor for rotations (ICC: −0.33–0.04, LOAs < 7.8°). The present study shows that a single inertial measurement unit (RSQ Motion, RSQ Technologies, Poznan, Poland) could be successfully used to assess the active hip range of motion in healthy subjects, comparable to other methods accuracy.

## 1. Introduction

An integral part of the clinical examination of the hip joint is the assessment of the range of motion (ROM). Determining any limitation allows for making a diagnosis, planning therapy, and objectifying the effects of treatment. In the literature, deficits of hip ROM have been reported in the case of osteoarthritis, femoroacetabular impingement syndrome, or osteitis pubis, and also as a risk factor for groin or knee injury [1,2,3,4]. The precision of the examination plays an important role; therefore, it is essential to use reliable measuring tools for this purpose.

Examination of the hip joint is hampered by its multiaxiality and compensatory movements of the spine or even the knee joint [5]. In previous studies, measurements of the range of motion in the hip joint were made using a visual estimation, goniometer, digital inclinometer, photographic assessment, smartphone application, or 3D motion capture systems [6,7,8,9]. Each method has been proven to be reliable in measuring the range of motion, but also has limitations. A major factor is always a subjective assessment of the examiner, and therefore carries the risk of error [10]. Examination using an inclinometer, goniometer, or smartphone is difficult due to the inability to stabilize the pelvis and take a measurement simultaneously, especially if performed by one tester. Previous studies have shown the impact of pelvic or even knee movement on hip ROM assessment outcomes [11,12]. In daily practice, photographic evaluation or advanced 3D motion capture is impractical, time-consuming, and expensive.

The inertial measurement unit (IMU) appears as an alternative measurement solution and is gaining popularity in the field of human motion capture. The device is made of a gyroscope, accelerometer, and magnetometer, and uses an appropriate algorithm, which allows for the calculation of precise motion measurement results [13]. The device is highly dependent on the algorithms used. The sensors themselves are subject to the risk of measurement errors resulting from, for example, time-variant sensor biases and measurement noise. These errors are limited by the constantly developing mathematical models, allowing precise measurement even from a reduced number of sensors [14,15].

In previous studies, IMU was validated to assess the range of motion of the shoulder joint, ankle joint, or cervical spine [16,17,18]. Their usefulness was also confirmed in the analysis of more complex activities such as walking or running, where one of the elements was the analysis of the kinematics of the hip joint [19,20]. However, in the studies published so far, we have yet to encounter the use of IMU in assessing a hip range of motion in clinical settings.

The small size of the device and the ease of assembly should allow for quick and precise measurement, especially when working with a patient one-on-one. The use of sensors should also allow for the elimination of compensatory movements of the pelvis by the possibility of its stabilization by the examiner. This is particularly important when measuring rotation, a crucial component of examining a patient with hip conditions [21]. In addition, if the sensor is attached to the body, it allows the user to assess the range of motion on his own. It also provides the opportunity to use it during more complex activities like sports or work. Considering the mentioned advantages of IMU, it is reasonable to check its usefulness in clinical conditions, which is the primary goal of this work.

Some widely available measurement systems on the market use IMUs. In a recent review, Garimella et al. point out that manufacturers use different computational algorithms in their devices, making them impossible to compare [22].

This study aimed to evaluate one of the IMU devices—the RSQ Motion (RSQ Technologies, Poznan, Poland) sensor—and check its reliability and validity against digital inclinometer and universal goniometer in measuring the active range of motion of the hip in healthy individuals, addressing the gap in clinical IMU usage.

## 2. Materials and Methods

### 2.1. Participants

Twenty healthy participants (10 female, 10 men; age: 27 ± 5 years; height: 174.4 ± 10.6 cm; body mass: 70.4 ± 13.6 kg) were recruited for the study based on inclusion criteria: no history of surgeries or injuries in the lower limb or spine, no pain in the hip or spine within the last 6 months, and no other diseases that may affect the test result. For confirmation of normal hip function, each of the participants completed the hip joint functional assessment questionnaire—Polish version of the Hip Disability and Osteoarthritis Outcome Score (HOOS PL) (mean score ± standard deviation: 99.7 ± 0.9) [23]. The study was approved by the Bioethical Committee of the University of Medical Sciences in Poznań, Poland (no. 13/21) and met the criteria of the Declaration of Helsinki. Before the study, all participants were informed about the purpose of the study and signed consent forms to participate in the study.

### 2.2. Instrumentation

Hip joint range of motion was measured using three devices:Universal plastic goniometer with a length of 20.32 cm, obtaining a measurement every 1° on a 360° scale.Baseline digital inclinometer (Fabrication Enterprises, White Plains, NY, USA). The manufacturer ensured a measurement accuracy of 0.5°.IMU by “RSQ Motion” (RSQ Technologies, Poznan, Poland). The RSQ Motion system is a class I medical device with a measuring function, manufactured under the RSQ Technologies brand, intended for the analysis of body movement. “RSQ Motion Sensor” performs measurements using MEMS sensors: accelerometer, gyroscope, and magnetometer. These data are sent as a digital electrical signal to a microcontroller, which processes it, filters it and calculates the orientation of the Motion Sensor and presents it in the form of quaternions. A quaternion is a mathematical construct with four dimensions resembling a complex number. “RSQ Motion” system incorporates a Madgwick algorithm for performing quaternion calculations. Details regarding the construction of the sensor and the mathematical procedures used for the measurement of orientation can be found in a previously published work [24]. The module with the microcontroller has a built-in Bluetooth low-energy antenna and communicates with the RSQ Motion Hub or an application using this protocol. The accuracy of the sensors was tested in laboratory conditions by comparing it to the Kuka robot (0.15°). The results showed excellent accuracy and repeatability, which allows us to believe that the device used will not affect the final results [25].

### 2.3. Study Protocol

The study was conducted by two independent testers, both experienced physiotherapists working clinically with orthopedic patients and skilled in assessing range of motion. Before the current study, a pilot trial was conducted on a group of 5 subjects to establish a uniform measurement protocol.

Every study participant was evaluated on two independent days, in the same place in the morning hours in room temperature. The subjects were wearing sports clothes and were without shoes. During the first day, two investigators performed an assessment (3 measurements for every direction of movement) with an hour interval to assess inter-rater reliability. After three days, another test was performed by one researcher to demonstrate intra-rater reliability.

Before the measurement to familiarize themselves with the measurements, all participants were asked to perform 2 test repetitions before each tested direction. The order of the tested movements was established according to the protocol: active flexion in the standing position, active external rotation, and internal rotation in the prone position.

#### 2.3.1. Active Hip Flexion in Standing Position

The participant stands against the wall with his feet in a relaxed position and supports himself with the opposite hand to the tested leg to maintain balance. During the examination, the patient is asked to lift the thigh with a flexed and relaxed knee as high as possible and maintain this position for about 1 s. The limb is held in the obtained position by the assistant for measurement with the goniometer and inclinometer (Figure 1). The examiner visually controls the occurrence of possible compensation—flexion of the other knee or lateral flexion of the trunk and corrects it if necessary. The measurement is performed 3 times for both limbs using three devices:IMU: The device is calibrated according to the protocol and zeroed at fixed vertical reference. The sensor is attached with a sticker 5 cm above the patella of the examined limb. When the maximum range of motion is reached, the recording is made by clicking the button on the so-called “clicker” and saved in a connected application.Inclinometer: The device is zeroed at a fixed vertical surface. The measurement is made by placing the inclinometer on the distal part of the thigh. The recording is made at the same time as IMU.Goniometer: The axis is placed at the height of the greater trochanter; one arm is directed along the femur, and the other vertically, parallel to the wall. The measurement is made right after the inclinometer and IMU.

#### 2.3.2. Active External and Internal Rotation in the Prone Position

Participant lying prone on the couch. To maintain the position of the limbs, a foam roller with a diameter of 13 cm was placed between the knees to avoid adduction/abduction movement of the hip. The knee of the test leg is bent to 90°. The assistant stabilizes the pelvis manually while the participant performs an active external rotation movement as far as possible. The obtained position is maintained by the assistant to take measurements. The same test is performed for internal rotation. The measurement is made 3 times for both limbs using three devices (Figure 2 and Figure 3):IMU: The device is calibrated according to the protocol and zeroed at a fixed vertical surface. The sensor is attached with a sticker 5 cm proximal to the ankle joint of the examined limb. When the maximum range of motion is reached, the recording is made by clicking the button on the so-called “clicker” and saved in a connected application.Inclinometer: The device is zeroed at a fixed vertical surface. The measurement is made by placing the inclinometer along the tibia underneath the IMU sensor. The recording is made at the same time as IMU.Goniometer: Axis of rotation set at the tibial tuberosity. Movable arm along the edge of the tibia; stationary arm set perpendicular to the couch.

### 2.4. Statistical Analysis

All statistical analyses were performed using the XLSTAT software (version: 2023 5.2.1413.0) program (Addinsoft Inc. New York, NY, USA). The Shapiro–Wilk test was used for the determination of the data distribution. Wilcoxon’s signed-rank test was used to compare continuous variables between the two groups. The values were presented as the mean ± standard deviation (minimal–maximal value). To assess the agreement of active hip ROM between IMU—RSQ Motion sensors, a goniometer, and an inclinometer for each movement, Intraclass Correlation Coefficient (ICC) and Bland–Altman analysis were used. Furthermore, inter-rater and intra-rater reliability were also assessed by using ICC and Bland–Altman analysis. Limits of agreement (LOA) were calculated using Bland–Altman plots. Intra- and inter-rater reliability was tested using a two-way random, single-measures, absolute agreement model ICC (2,1). Limits of agreement (LoA) are calculated as the mean of differences between two measurements ±1.96 × their standard deviation, using the Bland–Altman plot. We defined ICCs of less than 0.50 as indicative of poor reliability, values between 0.50 and 0.75 as indicative of moderate reliability, values between 0.75 and 0.90 as indicative of good reliability, and values greater than 0.90 as indicative of excellent reliability [26]. The precision of the individual measurements (the absolute reliability) was assessed with the Standard Error of Measurement (SEM). The sensitivity to change (the minimal amount of a change that a measurement must show to be greater than the within-subject variability and measurement error) was calculated as Minimal Detectable Change (MDC) with 95% confidence (MDC95) [27].

## 3. Results

### 3.1. Hip Flexion in Standing Position

#### 3.1.1. Concurrent Validity

The ICC analysis showed good correlation between IMU and inclinometer and goniometer. Limit of agreement (LOA) was below 9°. Correlation between goniometer and inclinometer was excellent with LOA below 2° (Table 1 and Figure 4).

#### 3.1.2. Inter-Rater Reliability

Inter-rater reliability was moderate for all tested instruments. LOA was the lowest for the goniometer—below 11.8°—and 12.8° and 22.4° for the inclinometer and IMU, respectively. SEM was below 1° and MDC95 was below 2.8° for all tested instruments (Table 2 and Figure 5).

#### 3.1.3. Intra-Rater Reliability

Intra-rater reliability for flexion was excellent for IMU and goniometer and good for inclinometer with an LOA below 6.7°. SEM was below 0.4° and MDC95 was below 1.1° for all tested instruments (Table 3 and Figure 6).

### 3.2. Hip Prone Internal Rotation

#### 3.2.1. Concurrent Validity for Hip Prone Internal Rotation

The ICC analysis showed an excellent correlation between IMU and inclinometer and goniometer. LOA was below 19.5° for tester 2 and below 3.7° for tester 1. Correlation between goniometer and inclinometer was also excellent with an LOA below 3.6° (Table 4 and Figure 7).

#### 3.2.2. Inter-Rater Reliability for Prone Internal Rotation

Inter-rater reliability for prone internal rotation was poor for all tested instruments with LOA below 33.6°. SEM was below 1.6° and MDC95 was below 4.3° for all tested instruments (Table 5 and Figure 8).

#### 3.2.3. Intra-Rater Reliability for Prone Internal Rotation

Intra-rater reliability for prone internal rotation was excellent for all tested instruments with LOA below 5.8°. SEM was below 0.3° and MDC95 was below 0.7° for all tested instruments (Table 6 and Figure 9).

### 3.3. Hip Prone External Rotation

#### 3.3.1. Concurrent Validity for Prone External Rotation

The ICC analysis showed a good–excellent correlation between IMU sensors and inclinometer and excellent correlation between IMU and goniometer. The limit of agreement (LOA) was below 7.8°. Correlation between goniometer and inclinometer was also excellent with LOA below 6.8° (Table 7 and Figure 10).

#### 3.3.2. Inter-Rater Reliability for Prone External Rotation

Inter-rater reliability for prone external rotation was poor for all tested instruments with LOA below 25.7°. SEM was below 1.2° and MDC95 was below 3.2° for all tested instruments (Table 8 and Figure 11).

#### 3.3.3. Intra-Rater Reliability for Prone External Rotation

Intra-rater reliability for prone external rotation was excellent for all tested instruments with LOA below 7°. SEM was below 0.3° and MDC95 was below 0.8° for all tested instruments (Table 9 and Figure 12).

## 4. Discussion

According to our knowledge, this is the first study to evaluate the validity and reliability of an IMU for active hip joint range of motion assessment in healthy subjects. The significant findings in the study are that the IMU showed mostly excellent validity compared with the “gold standard” universal goniometer and digital inclinometer. Regarding reliability, our results show a good–excellent correlation when the measurements are taken by the same examiner (intra-rater). When the study is performed by two examiners (inter-rater), the results were moderate for the flexion and poor for both external and internal. This reliability was inferior for every instrument used, likely associated with the assessment method and not with the IMU or other devices. We also found out that if reliability was excellent, then LOA was lower than 10°. Our results are consistent with those of Koegh et al., who reported ICC > 0.75, LOA < ±9.8°, and SEM < 5° as criteria for good validity for smartphone assessment [28]. Small values of SEM (<1.6°) and MDC95 (<4.3°) in our study indicate good absolute reliability.

It is difficult to compare our results with others due to the lack of studies examining the IMU in the assessment of the hip joint range of motion in the available literature. This is also the first study assessing active hip flexion in the standing position. We chose this position because it is used more often than lying on the back in everyday life. Moreover, isolating the flexion movement in the hip joint without additional pelvis and spine movement is difficult and could disturb the measurement [12].

Considering the tested position, the results of the research in which IMU was tested in gait analysis are the closest to our position [29,30]. Zugner et al. used inertial sensors to assess gait in people after hip arthroplasty and showed a good correlation with the optoelectric system for the hip flexion gait phase (left hip—ICC: 0.73; right hip—ICC: 0.75) [19]. It should be noted, however, that the range of flexion during gait is much smaller than that used in our study.

As mentioned, previous studies did not use IMU for the hip joint range of motion assessment. The most similar are the studies which conducted assessments via smartphone applications with built-in inertial units. Marshall et al. used inertial sensors located in the phone and compared them to the 3D motion analysis system [31]. They tested active flexion in the supine position with the sensor placed in front of the thigh as in our study. Compared to the 3D system, they showed good agreement (ICC: 0.81, SEM: 1.55°). The reported inter- and intra-reliability for the flexion range of motion in the mentioned study was lower than in our study, with ICCs of 0.52 and 0.59, respectively. A similar study was conducted by Charlton et al., who additionally used an analog inclinometer for measurement [32]. The main difference between our studies is that they tested the passive range of motion. Intra-tester reliability correlation was similar to our results for smartphone (ICC: 0.86; SEM: 2.3°) and inclinometer (ICC: 0.9; SEM: 2.8°).

Similarly to flexion, in many studies, prone rotation is assessed passively, which makes a comparison to an active one difficult [7,9,12]. Marshall et al. evaluated active rotation in the prone position and showed good agreement for external rotation and poor agreement for internal rotation compared to the 3D motion capture [31]. Ganokroj et al. also compared the smartphone application to a 3D system and showed a good correlation in both rotation directions; however, the passive position was tested [9].

Regarding inter-rater reliability mentioned, Marshall et al. reported ICCs of 0.95 and 0.94 for internal and external rotations, respectively, compared to our study that found ICCs of 0.02 for prone internal rotation and −0.29 for external rotation [31]. Intra-rater values were lower than ours, with ICCs of 0.44 for internal and 0.50 for external rotation. Their SEM and MDC values for internal rotation were 5.54° and 6.52°, respectively, which were also higher than ours: SEM, 0.3° and MDC, 0.7°. Similarly for external rotation, their SEM and MDC values were also higher—7.05° and 7.36°, respectively—compared to ours (SEM, 0.3° and MDC95, 0.8°) for IMU. Aefsky et al., in their work, used a similar method to assess rotation but with a standard goniometer [33]. Compared to our work, they also showed a higher level of inter-rater agreement (ICCs > 0.9) and a similar level of intra-rater agreement: 0.87–0.95 versus 0.96–0.99 in our study.

Similarly, higher inter-rater ICCs were reported by Ganokroj et al., at 0.81 and 0.76 for prone internal and external rotations, with SEM values of 4.1° and 3.9°, respectively [9]. Again, intra-rater agreement was close to our work: ICCs of 0.90 with SEM 2.8 for external rotation and 0.91 with SEM 3.22° for internal rotation. The LOA was similar to ours (<10°).

The goniometer, inclinometer, and IMU showed similar levels of reliability. Considering this, the low level of inter-rater reliability, especially for prone rotation, may potentially result from technical errors during measurement, not from the precision of the tools used. One of the possible causes may be the compensatory movement of the pelvis and its position itself, which may distort the result. Previous studies have shown that an increased pelvis anteversion may reduce the possibility of flexion or rotation [34]. Nussbaumer et al. showed that the goniometer examination of the range of flexion and rotation is subject to error mainly due to additional pelvic movement [12]. In our study, we wanted the tested positions to be as close as possible to everyday clinical practice. Therefore, during the prone rotation assessment, the tester stabilized the pelvis manually, which is associated with the risk of differences in the pressure on the pelvis and the determination of the moment of its movement, which may occur between assessors. Although our preliminary results showed no differences between the testers regarding the mean values of the range of movement tested, other researchers indicate that the testers’ experience may have an impact on the hip assessment [35].

Interestingly, we achieved significantly better levels of inter-rater agreement in standing active flexion. During flexion, it is reported that up to 1/3 of the movement may come from the pelvic movement, which is one of the reasons why we decided on such an examination position for the assessment of the entire functional range of motion [36]. Therefore, it seems reasonable to believe that the better inter-rater reliability values for flexion than rotation resulted from the lack of additional pelvis stabilization. Another issue is the position of the research tool during the measurement. Despite the established study protocol, researchers may consider reference points differently, which may also result from the individual constitution of participants. In the case of a goniometer, it may not be easy to set one of the arms on the vertical axis, especially since it was not equipped with spirit levels. The placement of the inclinometer along the long axis of the tibia or thigh is also to some extent based on the subjective assessment of the examiner. During the measurement using the IMU, errors may also result from subtle differences in adhering the device and additional skin movements during active motion, which also occur with other systems, such as the 3D motion capture [37].

The examination of the range of motion is one of the basic elements of clinical assessment in orthopedics or physiotherapy. In clinical practice, the most commonly used measurement tools are the goniometer or the inclinometer. Both tools need another person for pelvic stabilization while assessing pure hip joint movement. If performed alone, taking a precise measurement is much more challenging. The solution to this problem may be inertial sensors such as RSQ Motion, which can be easily mounted on the examined limb and allow pelvis stabilization simultaneously. The small size of the device and the ease of assembly also allow it to be used at home by the patient. This creates the opportunity for a better monitoring of progress, as well as compliance with home exercise. However, the reliability of the measurement performed by the patient requires further research. The proposed study protocol with IMU may be used in future studies to assess the range of motion or the proprioception of patients with hip complaints or after hip surgery.

Like any study, ours also has limitations. Firstly, we examined young and healthy people with possibly more flexibility in the spine and hip; therefore, the results cannot be transferred to people with hip conditions or in older age and without symptoms. Yet, the methods of evaluation would not be different in clinical scenarios. Secondly, the results may be affected by other movements of the surrounding joints; therefore, it seems reasonable to use two sensors in the future, one of which would be located, for example, on the pelvis. Another problem may be the issue of artifacts related to the movement of the skin in the case of glued sensors, as well as anatomical differences of the subjects [35]. To minimize this, we placed the stickers on an area with limited adipose tissue at precisely defined points. Thirdly, the tested range of motion was active and subjectively determined by the participant; therefore, the results cannot be directly related to the more frequently used passive assessment in clinical practice. Fourthly, due to the study protocol, we cannot exclude the influence of physical and mental fatigue of the subjects on the results.

## 5. Conclusions

The use of a single inertial measurement unit (IMU, represented by RSQ Motion) in assessing active hip motion is valid compared to the universal goniometer and digital inclinometer. All instruments (including IMU) showed comparable and mostly good-to-excellent reliability for both at two different sessions of assessment and between both testers except for interrater reliability of prone internal and external rotation (regardless of the device). Due to better access, relatively low cost, and ease of use, inertial systems could be increasingly used in the field of orthopedics and physical therapy. More improvements could be anticipated with two or more sensors used to depict a hip range of motion without possible compensations and the need for specialist evaluation.

## Figures and Tables

**Figure 1 sensors-23-08782-f001:**
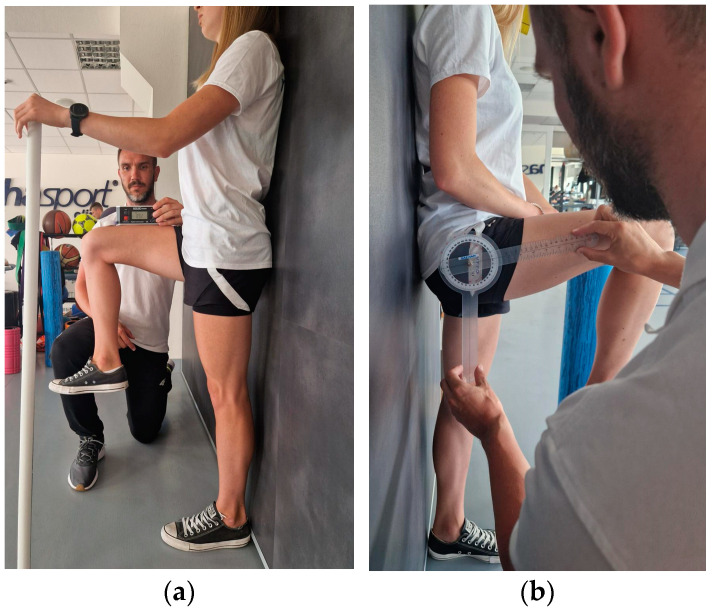
Active flexion range of motion assessment with (**a**) IMU and inclinometer and (**b**) goniometer.

**Figure 2 sensors-23-08782-f002:**
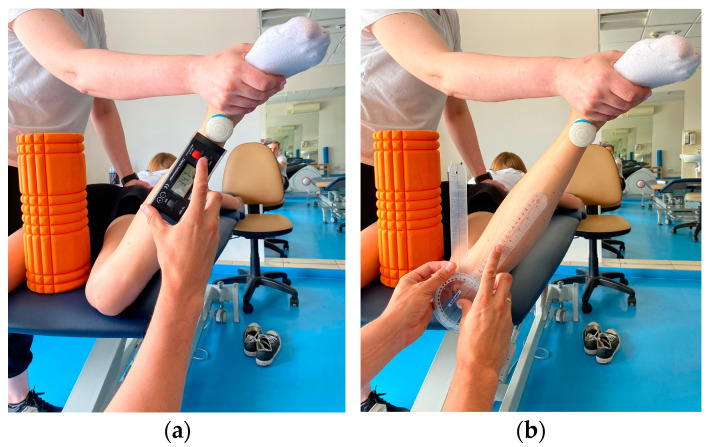
Active internal rotation range of motion assessment with (**a**) IMU and inclinometer and (**b**) goniometer.

**Figure 3 sensors-23-08782-f003:**
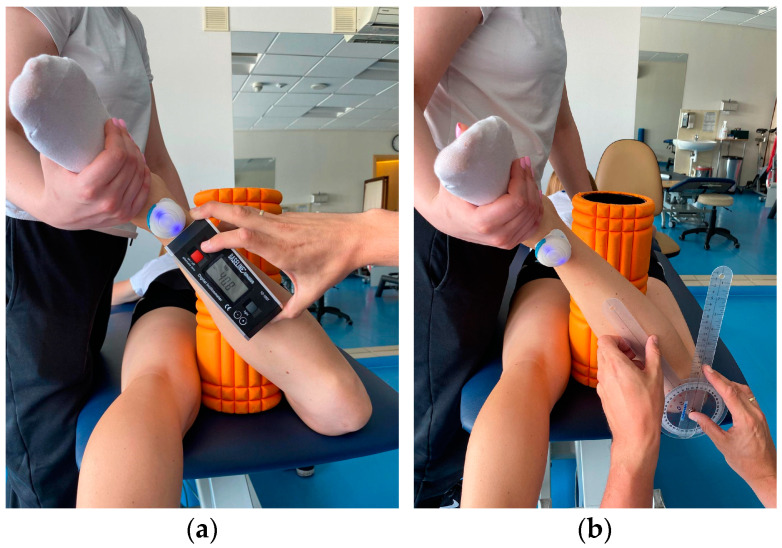
Active external rotation range of motion assessment with (**a**) IMU and inclinometer and (**b**) goniometer.

**Figure 4 sensors-23-08782-f004:**
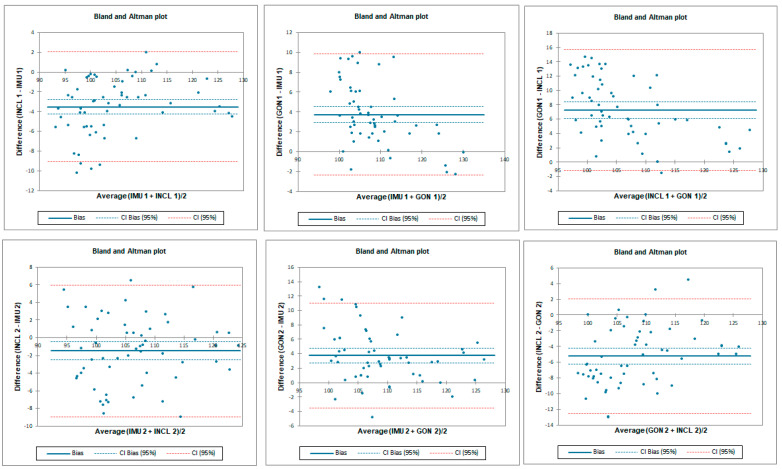
Bland–Altman plots for hip flexion in standing position for tester 1 (upper plots) and tester 2 (lower plots)—concurrent validity. IMU 1—Inertial measurement unit used by tester 1; IMU 2—Inertial measurement unit used by tester 2; GON 1—goniometer used by tester 1; GON 2—goniometer used by tester 2; INCL 1—inclinometer used by tester 1; INCL 2—inclinometer used by tester 2. The *y*-axis represents the difference between the methods and the *x*-axis represents average of measurements.

**Figure 5 sensors-23-08782-f005:**
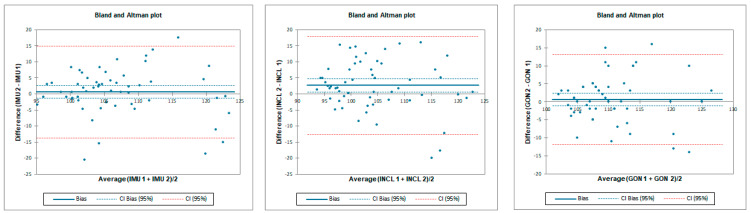
Bland–Altman plots for hip flexion in standing position—inter-rater reliability. IMU 1—Inertial measurement unit used by tester 1; IMU 2—Inertial measurement unit used by tester 2; GON 1—goniometer used by tester 1; GON 2—goniometer used by tester 2; INCL 1—inclinometer used by tester 1; INCL 2—inclinometer used by tester 2. The *y*-axis represents the difference between the testers and the *x*-axis represents average of measurements.

**Figure 6 sensors-23-08782-f006:**
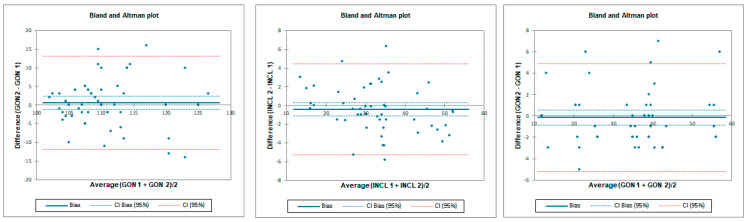
Bland–Altman plots for hip flexion in standing position—intra-rater reliability. IMU 1—Inertial measurement unit results from day 1; IMU 2—Inertial measurement unit results from day 2; GON 1—goniometer results from day 1; GON 2—goniometer results from day 2; INCL 1—inclinometer results from day 1; INCL 2—inclinometer results from day 2. The *y*-axis represents the difference between results from day 1 and day 2 and the *x*-axis represents average of measurements.

**Figure 7 sensors-23-08782-f007:**
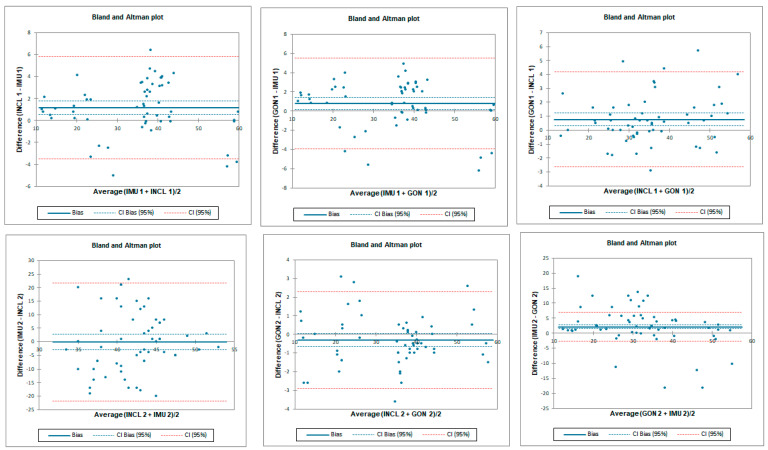
Bland–Altman plots for hip prone internal rotation for tester 1 (upper plots) and tester 2 (lower plots)—concurrent validity. IMU 1—Inertial measurement unit used by tester 1; IMU 2—Inertial measurement unit used by tester 2; GON 1—goniometer used by tester 1; GON 2—goniometer used by tester 2; INCL 1—inclinometer used by tester 1; INCL 2—inclinometer used by tester 2. The *y*-axis represents the difference between the methods and the *x*-axis represents average of measurements.

**Figure 8 sensors-23-08782-f008:**
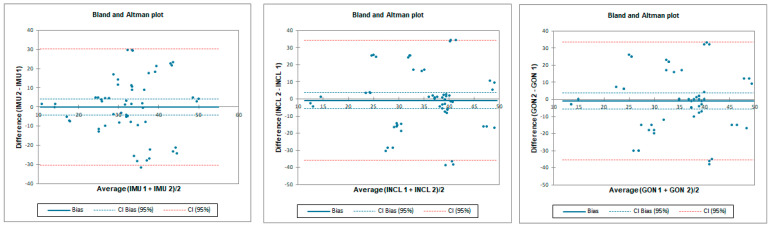
Bland–Altman plots for hip prone internal rotation—inter-rater reliability. IMU 1—Inertial measurement unit used by tester 1; IMU 2—Inertial measurement unit used by tester 2; GON 1—goniometer used by tester 1; GON 2—goniometer used by tester 2; INCL 1—inclinometer used by tester 1; INCL 2—inclinometer used by tester 2. The y-axis represents difference between the testers and the x-axis represents average of measurements.

**Figure 9 sensors-23-08782-f009:**
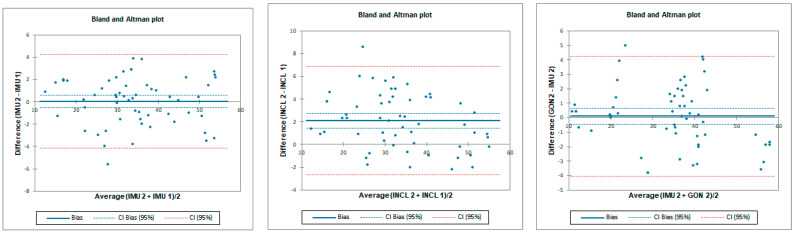
Bland–Altman plots for hip prone internal rotation intra-rater reliability. IMU 1—Inertial measurement unit results from day 1; IMU 2—Inertial measurement unit results from day 2; GON 1—goniometer results from day 1; GON 2—goniometer results from day 2; INCL 1—inclinometer results from day 1; INCL 2—inclinometer results from day 2. The *y*-axis represents the difference between results from day 1 and day 2 and the *x*-axis represents average of measurements.

**Figure 10 sensors-23-08782-f010:**
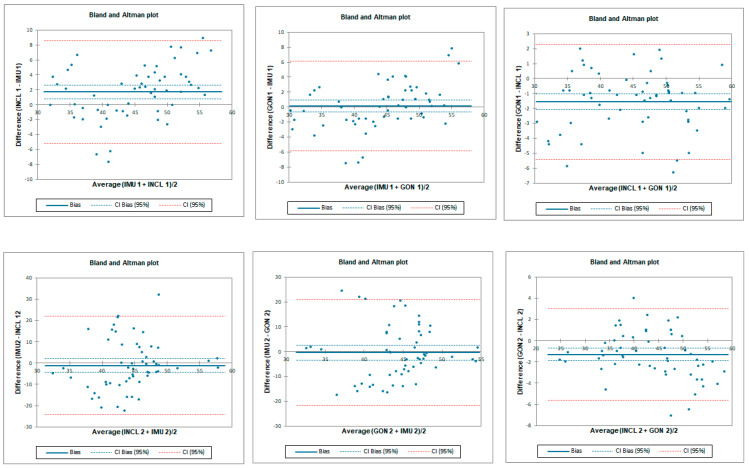
Bland–Altman plots for hip prone external rotation for tester 1 (upper plots) and tester 2 (lower plots)—concurrent validity. IMU 1—Inertial measurement unit used by tester 1; IMU 2—Inertial measurement unit used by tester 2; GON 1—goniometer used by tester 1; GON 2—goniometer used by tester 2; INCL 1—inclinometer used by tester 1; INCL 2—inclinometer used by tester 2. The *y*-axis represents the difference between the methods and the *x*-axis represents average of measurements.

**Figure 11 sensors-23-08782-f011:**
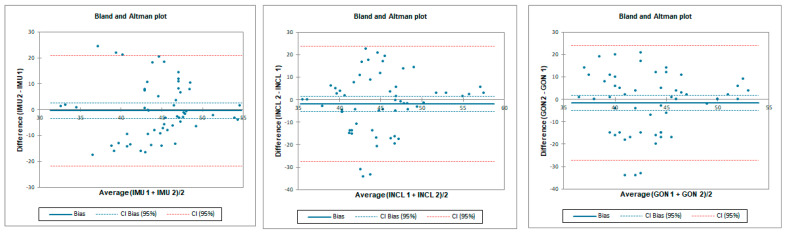
Bland–Altman plots for hip prone external rotation—inter-rater reliability. IMU 1—Inertial measurement unit used by tester 1; IMU 2—Inertial measurement unit used by tester 2; GON 1—goniometer used by tester 1; GON 2—goniometer used by tester 2; INCL 1—inclinometer used by tester 1; INCL 2—inclinometer used by tester 2. The *y*-axis represents the difference between the testers and the *x*-axis represents average of measurements.

**Figure 12 sensors-23-08782-f012:**
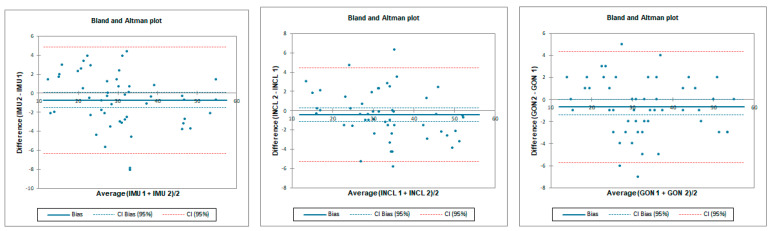
Bland–Altman plots for hip prone external rotation—intra-rater reliability. IMU 1—Inertial measurement unit results from day 1; IMU 2—Inertial measurement unit results from day 2; GON 1—goniometer results from day 1; GON 2—goniometer results from day 2; INCL 1—inclinometer results from day 1; INCL 2—inclinometer results from day 2. The *y*-axis represents the difference between results from day 1 and day 2 and the *x*-axis represents average of measurements.

**Table 1 sensors-23-08782-t001:** Intra Class Coefficient with 95% confidence interval, bias and LOA between the data gathered using different instruments by testers 1 and 2 for flexion assessment.

Tester 1				Tester 2		
	ICC	Bias (°)	LOA (°)	ICC	Bias (°)	LOA (°)
IMU vs. INCL	0.88	3.3 ± 2.9	−2.4; 9.0	0.90	1.3 ± 3.2	−5.0; 7.6
IMU vs. GON	0.90	−3.5 ± 3.2	−9.8; 2.8	0.91	−3.8 ± 3.7	−11.1; 3.4
INCL vs. GON	0.94	−6.8 ± 4.1	−14.8; 1.3	0.93	−5.1 ± 3.6	−12.22; 2.0

Abbreviations: IMU—Inertial measurement unit; INCL—digital inclinometer; GON—universal goniometer; ICC—Intra Class Correlation Coefficient; 95% CI—95% confidence interval; LOA—Limits of Agreement.

**Table 2 sensors-23-08782-t002:** Inter-rater reliability indicators in hip flexion for all of the measuring devices; tester 1 vs. tester 2.

Flexion	Tester 1 Mean ± SD (Min–Max)	Tester 2 Mean ± SD (Min–Max)	ICC	SEM (°)	MDC95 (%) (°)	Bias (°) ±SE	LOA (°)
IMU	107.4 ± 8.2 (91.1–130.1)	107.1 ± 7.8 (89.3–129.6)	0.58	1.0	2.8	1.0 ± 10.9	−20.4; 22.4
Inclinometer	104.0 ± 8.8 (90.4–128.9)	105.8 ± 7.9 (93.0–127.8)	0.59	0.7	1.9	−1.8 ± 7.4	−16.4; 12.8
Goniometer	110.9 ± 6.7 (101.0–134.0)	111.0 ± 6.2 (100.0–128.0)	0.58	0.6	1.6	−0.2 ± 6.1	−12.7; 11.8

Abbreviations: SD—Standard Deviation; ICC—Intra Class Correlation Coefficient; 95% CI—95% confidence interval; SEM—Standard Error of Measurement; MDC—Minimal Detectable Change with 95% confidence; SE—Standard Error; LOA—Limits of Agreement.

**Table 3 sensors-23-08782-t003:** Intra-rater reliability indicators in hip flexion for all of measuring devices.

Flexion	Test Mean ± SD (Min–Max)	Retest Mean ± SD (Min–Max)	ICC	SEM (°)	MDC95 (%) (°)	Bias (°) ±SE	LOA (°)
IMU	106.0 ± 7.2 (89.3–129.6)	106.8 ± 7.7 (89.2–133.8)	0.94	0.4	1.1	−1.4 ± 4.1	−9.6; 6.7
Inclinometer	104.3 ± 6.9 (93.0–127.8)	105.8 ± 7.5 (93.0–138.0)	0.87	0.4	1.1	−2.0 ± 4.3	−10.5; 6.5
Goniometer	109.8 ± 5.6 (100.0–128.0)	110.1 ± 6.0 (100.0–131.0)	0.91	0.3	0.9	−0.9 ± 3.5	−7.7; 5.9

Abbreviations: SD—Standard Deviation; ICC—Intra Class Correlation Coefficient; 95% CI—95% confidence interval; SEM—Standard Error of Measurement; MDC—Minimal Detectable Change; SE—Standard Error; LOA—Limits of Agreement.

**Table 4 sensors-23-08782-t004:** Intra Class Coefficient 95% CI—95% confidence interval, bias and LOA between the data gathered using different instruments by testers 1 and 2 for internal rotation assessment.

Tester 1				Tester 2		
	ICC	Bias (°)	LOA (°)	ICC	Bias (°)	LOA (°)
IMU vs. INCL	0.97	−1.7 ± 2.6	−6.8; 3.3	0.99	−0.3 ± 10.1	−20.1; 19.5
IMU vs. GON	0.99	−1.9 ± 2.8	−7.6; 3.7	0.97	−0.2 ± 10.0	−19.8; 19.4
INCL vs. GON	0.99	−0.2 ± 1.6	−3.3; 2.8	0.98	0.1 ± 1.7	−3.3; 3.6

Abbreviations: IMU—Inertial measurement unit; INCL—digital inclinometer; GON—universal goniometer; ICC—Intra Class Correlation Coefficient; 95% CI—95% confidence interval; LOA—Limits of Agreement.

**Table 5 sensors-23-08782-t005:** Inter-rater reliability indicators in hip prone internal rotation for all of the measuring devices; Operator 1 vs. Operator 2.

Internal Rotation	Tester 1 Mean ± SD (Min–Max) (°)	Tester 2 Mean ± SD (Min–Max) (°)	ICC	SEM (°)	MDC_95_ (%) (°)	Bias (°) ±SE	LOA (°)
IMU	33.4 ± 12.0 (10.1–61.4)	33.3 ± 12.1 (10.6–58.9)	0.02	1.6	4.3	0.1 ± 16.8	−32.8; 33.6
Inclinometer	35.2 ± 11.5 (12.0–60.2)	34.2 ± 11.5 (10.8–58.5)	0.04	1.5	4.1	0.6 ± 15.9	−30.5; 31.7
Goniometer	35.4 ± 11.4 (12.0–60.0)	34.4 ± 11.5 (11.0–57.0)	0.04	1.5	4.1	0.9 ± 15.8	−30.0; 31.2

Abbreviations: SD—Standard Deviation; ICC—Intra Class Correlation Coefficient; 95% CI—95% confidence interval; SEM—Standard Error of Measurement; MDC—Minimal Detectable Change; SE—Standard Error; LOA—Limits of Agreement.

**Table 6 sensors-23-08782-t006:** Intra-rater reliability indicators in hip prone internal rotation for all of the measuring devices.

Internal Rotation	Test Mean ± SD (Min–Max) (°)	Retest Mean ± SD (Min–Max) (°)	ICC	SEM (°)	MDC_95_ (%) (°)	Bias ± SE	LOA (°)
IMU	32.4 ± 12.3 (10.6–58.9)	32.0 ± 11.9 (10.2–60.7)	0.96	0.3	0.7	0.3 ± 2.8	−5.1; 5.8
Inclinometer	33.6 ± 11.5 (10.8–58.5)	33.2 ± 11.3 (11.9–59.0)	0.99	0.2	0.6	0.3 ± 2.5	−4.6; 5.2
Goniometer	33.6 ± 11.6 (11.0–57.0)	33.2 ± 11.5 (12.0–60.0)	0.99	0.2	0.6	0.3 ± 2.5	−4.6; 5.2

Abbreviations: SD—Standard Deviation; ICC—Intra Class Correlation Coefficient; 95% CI—95% confidence interval; SEM—Standard Error of Measurement; MDC—Minimal Detectable Change; SE—Standard Error; LOA—Limits of Agreement.

**Table 7 sensors-23-08782-t007:** Intra Class Coefficient with 95% confidence interval, bias and LOA between the data gathered using different instruments by testers 1 and 2 for external rotation assessment.

Tester 1				Tester 2		
	ICC	Bias (°)	LOA (°)	ICC	Bias (°)	LOA (°)
IMU vs. INCL	0.88	−1.0 ± 3.3	−7.5; 5.4	0.95	−0.4 ± 3.2	−6.6; 5.8
IMU vs. GON	0.97	1.1 ± 3.4	−5.6; 7.8	0.99	1.1 ± 2.6	−4.0; 6.2
INCL vs. GON	0.99	2.1 ± 2.4	−2.5; 6.8	0.96	1.5 ± 2.4	−3.2; 6.2

Abbreviations: IMU—Inertial Measurement Unit; INCL—digital inclinometer; GON—universal goniometer; ICC—Intra Class Correlation Coefficient; 95% CI—95% confidence interval; LOA—Limits of Agreement.

**Table 8 sensors-23-08782-t008:** Inter-rater reliability indicators in hip prone external rotation for all of the measuring devices; tester 1 vs. tester 2.

External Rotation	Tester 1 Mean ± SD (Min–Max) (°)	Tester 2 Mean ± SD (Min–Max) (°)	ICC	SEM (°)	MDC_95_ (%) (°)	Bias ± SE	LOA (°)
IMU	44.4 ± 6.6 (25.0–56.3)	43.6 ± 7.8 (24.1–55.4)	−0.29	1.1	3.0	0.8 ± 11.5	−21.8; 23.4
Inclinometer	45.5 ± 7.3 (30.0–60.4)	44.0 ± 8.4 (25.8–64.7)	−0.24	1.2	3.2	1.4 ± 11.4	−22.8; 25.7
Goniometer	43.3 ± 7.3 (25.0–59.0)	42.5 ± 7.6 (24.0–57.0)	−0.33	1.1	3.0	0.8 ± 12.1	−22.8; 24.4

Abbreviations: SD—Standard Deviation; ICC—Intra Class Correlation coefficient; 95% CI—95% confidence interval; SEM—Standard Error of Measurement; MDC—Minimal Detectable Change; SE—Standard Error; LOA—Limits of Agreement.

**Table 9 sensors-23-08782-t009:** Intra-rater reliability indicators in hip prone external rotation for all of the measuring devices.

External Rotation	Test Mean ± SD (Min–Max) (°)	Retest Mean ± SD (Min–Max) (°)	ICC	SEM (°)	MDC_95_ (%) (°)	Bias ± SE	LOA (°)
IMU	44.5 ± 6.9 (27.7–55.4)	44.8 ± 7.2 (30.9–60.1)	0.98	0.3	0.8	−0.2 ± 3.1	−6.3; 5.8
Inclinometer	44.9 ± 7.7 (29.4–64.7)	44.3 ± 7.6 (27.3–50.2)	0.95	0.3	0.8	0.7 ± 3.2	−5.6; 7.0
Goniometer	43.3 ± 6.9 (27.0–57.0)	3.7 ± 7.5 (26.0–58.0)	0.97	0.2	0.7	−0.3 ± 2.6	−5.3; 4.8

Abbreviations: SD—Standard Deviation; ICC—Intra Class Correlation Coefficient; 95% CI—95% confidence interval; SEM—Standard Error of Measurement; MDC—Minimal Detectable Change; SE—Standard Error; LOA—Limits of Agreement.

## Data Availability

Not applicable.

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
