# Peer review of "Validity and Reliability of Inertial Measurement Units in Active Range of Motion Assessment in the Hip Joint"

_sensors, 2023, doi:10.3390/s23218782_

Round 1
Reviewer 1 Report
This study aimed to evaluate one of the IMU devices - the RSQ Motion sensor and check its reliability and validity against digital inclinometer and universal goniometer in measuring the active range of motion of the hip in healthy individuals. The paper is well-written and organized. The contribution of the paper is appropriate. Nevertheless, specific improvements are necessary before considering its publication, as outlined in the following comments.
1. In the abstract, the focus and conciseness of the obtained results should be enhanced.
2. The introduction section should emphasize the paper's contribution prominently.
3. Define all abbreviations upon their first usage, including RSQ.
4. Remove "Figure..." from heading and subheading titles, reserving it for explanations or relevant thematic sections.
5. Revise figure captions as "(a).....................and (b)....................."
6. Section 2.4 should incorporate Bland-Altman plots as plots.
7. Further discussion and interpretation are needed for the tables in the results section.
8. The paper could benefit from the inclusion of mathematical models within the scope of the search.
9. A comparison with previous scholarly findings is necessary to validate the presented work.
1. Many references appear outdated. Updating them to more recent sources is recommended.
None
Author Response
Author's Reply to the Review Report (Reviewer 1) – manuscript ID sensors-2577228
Title: Validity and reliability of inertial measurement units in active range
of motion assessment in the hip joint.
Authors: Łukasz Stołowski *, Maciej Niedziela, Bartłomiej Lubiatowski, Przemysław Lubiatowski, Tomasz Piontek
We thank you very kindly for all your comments and suggestions. They helped improve our article a lot. Below are our responses:
Review Report (Reviewer 1):
In the abstract, the focus and conciseness of the obtained results should be enhanced.
Our response: we made results more concise as suggested.
The introduction section should emphasize the paper's contribution prominently.
Our response: in the introduction section, we highlighted the challenges of assessing the hip joint range of motion, the limitations of existing measurement methods, and the potential advantages of using Inertial Measurement Units (IMUs) in clinical settings. We modified it to make the contribution of our work more prominent, as suggested.
Define all abbreviations upon their first usage, including RSQ.
Our response: „RSQ” this is the name of a technological company that provides used sensors, not an abbreviation itself but we add information about it during first usage.
Remove "Figure..." from heading and subheading titles, reserving it for explanations or relevant thematic sections.
Our response: we have changed it as suggested.
Revise figure captions as "(a).....................and (b)....................."
Our response: we have changed it as suggested.
Section 2.4 should incorporate Bland-Altman plots as plots.
Our response: We are still determining whether this applies to paragraph 2.4 - Statistical analysis or results. However, due to the large number of plots (36), we propose to leave them out of the article for clarity. If they are needed, we suggest including them in the supplement.
Further discussion and interpretation are needed for the tables in the results section.
Our response: Thank you for this comment. From our point of view, the results section is only about describing obtained results without any additional information. On the other hand, in the discussion section, we thoroughly interpreted our data, enriching them with our observations and conclusions from the conduction of the study. That is why we would like to stay with the current version of the results section.
The paper could benefit from the inclusion of mathematical models within the scope of the search.
Our response: This is a validation study in clinical environment, so that we were concentrated on the results of our examination but we’ve added information how mathematical models are important for precise measurements made by IMU and backed it up with references - lines:62-66. We consulted it with engineer who is expert in this field and that’s why we’ve added him as a co-author (Bartłomiej Lubiatowski) of the revised version of the manuscript.
A comparison with previous scholarly findings is necessary to validate the presented work.
Our response: There is little literature on this topic, but we compared our results with the most recent articles in the discussion section - lines: 343-388
Many references appear outdated. Updating them to more recent sources is recommended.
Our response: We’ve updated the references where it was possible for the most recent ones.
Reviewer 2 Report
This paper focuses on human subjects and requires participants to perform three hip joint activities.Three different measurement tools are utilized to assess hip joint range of motion. It compares the results between IMU and the other two measurement tools to validate the accuracy and reliability of a single IMU (RSQ Motion) in evaluating the hip joint activity range in healthy participants. In summary, this is primarily an experimental confirmatory study with practical significance. The validation process is comprehensive, but there are some minor questions to address before publication.
(1)Why did the author choose 20 healthy participants(10 female, 10 men; age: 27 ± 5 years; height: 174.4 ± 10.6 cm; 80 body mass: 70.4 ± 13.6 kg) as the study subjects? Moreover, the relatively small sample size may impact the general applicability of the research results.
(2)Concerning uncontrolled external factors, further error analysis should be considered. For instance, the substantial difference in clinical experience between measurement personnel (12 years vs. 5 years) mentioned in the paper could introduce measurement errors. It would be helpful to assess how much this might affect the research outcomes.
(3)Please provide detailed information about the timing of measurements, such as the specific time of day when measurements were taken, and consider the potential influence of fatigue effects. This would contribute to the credibility of the study results.
(4)Please provide a more detailed explanation of how ICC and Bland-Altman analyses were conducted. Offering additional information about these analytical methods would enhance the transparency of the research.
(5)Lines 317-320, 371-374:Please provide a detailed explanation of why the mentioned instances of larger experimental result errors are attributed to the assessment methods rather than the assessment equipment.
I also have some minor suggestions for the English. Generally speaking, the writing is good. However, there are some places where the sentences are difficult to understand.
Author Response
Author's Reply to the Review Report (Reviewer 2) – manuscript ID sensors-2577228
Title: Validity and reliability of inertial measurement units in active range
of motion assessment in the hip joint.
Authors: Łukasz Stołowski *, Maciej Niedziela, Bartłomiej Lubiatowski, Przemysław Lubiatowski, Tomasz Piontek
We thank you very kindly for all your comments and suggestions. They helped improve our article a lot. Below are our responses:
Review Report (Reviewer 2):
(1)Why did the author choose 20 healthy participants(10 female, 10 men; age: 27 ± 5 years; height: 174.4 ± 10.6 cm; 80 body mass: 70.4 ± 13.6 kg) as the study subjects? Moreover, the relatively small sample size may impact the general applicability of the research results.
Our response: The sample size calculation showed that with a power of 80% (2-sided testing at a significance level of 0.05) a sample size of 8 participants was needed to show a difference between two independent testers or between the devices. In the group of 20 participants, the test power was large (95%) [ref].
ref: Suresh K, Chandrashekara S. Sample size estimation and power analysis for clinical research studies. J Hum Reprod Sci. 2015;8(3):186. doi: 10.4103/0974-1208.165154.
- Concerning uncontrolled external factors, further error analysis should be considered. For instance, the substantial difference in clinical experience between measurement personnel (12 years vs. 5 years) mentioned in the paper could introduce measurement errors. It would be helpful to assess how much this might affect the research outcomes.
Our response: In our work, we wanted to focus on the validation of the device rather than comparing results dependent on the experience of the researchers. In the preliminary statistical results, the difference between the two testers' mean results was insignificant (p > 0.05). However, we removed it from the tables for clarity and to draw attention to the validation-related results. We added information about it in lines: 401-403.
- Please provide detailed information about the timing of measurements, such as the specific time of day when measurements were taken, and consider the potential influence of fatigue effects. This would contribute to the credibility of the study results.
Our response: We added missing information in study protocol and consider fatigue effects in a study limitations section.
- Please provide a more detailed explanation of how ICC and Bland-Altman analyses were conducted. Offering additional information about these analytical methods would enhance the transparency of the research.
Our response: We added missing information in a 2.4 paragraph - lines 222-226.
- Lines 317-320, 371-374:Please provide a detailed explanation of why the mentioned instances of larger experimental result errors are attributed to the assessment methods rather than the assessment equipment.
Our response: The mentioned errors mainly concern the results of prone rotation, where the issue of isolating the movement in the hip joint from the pelvis plays an important role. For this purpose, the examiner must stabilize the pelvis with his hand, which causes the risk of error resulting from the subjective determination of the moment of its movement. In the case of flexion testing, where we did not consider pelvic stabilization, we obtained better results. Considering this and the similar reliability level obtained for all devices, we concluded that the measurement method, not the equipment itself, is responsible for large errors. We described it in detail in the Discussion section in lines 382-410.
I also have some minor suggestions for the English. Generally speaking, the writing is good. However, there are some places where the sentences are difficult to understand.
Our response: We have checked it once again and improved.
Reviewer 3 Report
Thank you for allowing me to review this article, ,it is a topic of great interest as more clinicians and researchers are moving toward the use of IMU devices.
A couple minor things, you state on line 115 that participants were without shoes for the protocol but figure 1 has the participant in shoes. Very minor and not a hold up on publication to me.
second, line 413 needs clarification.
Author Response
Author's Reply to the Review Report (Reviewer 3) – manuscript ID sensors-2577228
Title: Validity and reliability of inertial measurement units in active range
of motion assessment in the hip joint.
Authors: Łukasz Stołowski *, Maciej Niedziela, Bartłomiej Lubiatowski, Przemysław Lubiatowski, Tomasz Piontek
We thank you very kindly for all your comments and suggestions. They helped improve our article a lot. Below are our responses:
Review Report (Reviewer 3):
A couple minor things, you state on line 115 that participants were without shoes for the protocol but figure 1 has the participant in shoes. Very minor and not a hold up on publication to me.
Our response: We agree. This is our oversight. The examination is carried out without shoes, and the photo is only for illustration.
second, line 413 needs clarification.
Our response: We clarified that as suggested.
Round 2
Reviewer 1 Report
The authors have not made revisions to the paper in line with all of my comments. The comments from the first round remain unsatisfactory. Therefore, the author should make the necessary revisions to the paper accordingly.
1. Section 2.4 should incorporate Bland-Altman plots as plots.
2. The paper could benefit from the inclusion of mathematical models within the scope of the search.
3. A comparison with previous scholarly findings is necessary to validate the presented work.
None
Author Response
We thank you very kindly for all your comments and suggestions. Below are our responses:
Section 2.4 should incorporate Bland-Altman plots as plots.
Our response: We included Bland-Altman plots as suggested.
The paper could benefit from the inclusion of mathematical models within the scope of the search.
Our response: We added information about the mathematical algorithm used in the IMU system and backed it up with references with further information about the mathematical models used. - lines 112-116 / reference: 24
A comparison with previous scholarly findings is necessary to validate the presented work.
Our response: As mentioned in the article, our work is the first to validate IMU sensors in assessing active hip range of motion in clinical setting. We compared and related our results to previous scientific reports:
- Validation of IMU sensors with the Kuka robot - line 114 / reference 24.
- Validation of IMU sensors to assess the range of motion of other body parts - line 56 / references: 16-18.
- Studies using other measuring tools for hip joint range of motion, including the gold standard such as the goniometer or optoelectric devices and most similar to our device smarthphone assesment where inertial units are build in - lines 349-388 / references: 28-32.